# Assessment of Narrow-Band Imaging Algorithm for Video Capsule Endoscopy Based on Decorrelated Color Space for Esophageal Cancer: Part II, Detection and Classification of Esophageal Cancer

**DOI:** 10.3390/cancers16030572

**Published:** 2024-01-29

**Authors:** Yu-Jen Fang, Chien-Wei Huang, Riya Karmakar, Arvind Mukundan, Yu-Ming Tsao, Kai-Yao Yang, Hsiang-Chen Wang

**Affiliations:** 1Department of Internal Medicine, National Taiwan University Hospital, Yun-Lin Branch, No. 579, Sec. 2, Yunlin Rd., Dou-Liu 64041, Taiwan; toby851072@gmail.com; 2Department of Internal Medicine, National Taiwan University College of Medicine, No. 1, Jen Ai Rd., Sec. 1, Taipei 10051, Taiwan; 3Department of Gastroenterology, Kaohsiung Armed Forces General Hospital, 2, Zhongzheng 1st Rd., Lingya District, Kaohsiung 80284, Taiwan; forevershiningfy@yahoo.com.tw; 4Department of Nursing, Tajen University, 20, Weixin Rd., Yanpu Township, Pingtung County 90741, Taiwan; 5Department of Mechanical Engineering, National Chung Cheng University, 168, University Rd., Min Hsiung, Chia Yi 62102, Taiwan; karmakarriya345@gmail.com (R.K.); d09420003@ccu.edu.tw (A.M.); d09420002@ccu.edu.tw (Y.-M.T.); 6Department of Medical Research, Dalin Tzu Chi Hospital, Buddhist Tzu Chi Medical Foundation, No. 2, Minsheng Road, Dalin, Chia Yi 62247, Taiwan; 7Hitspectra Intelligent Technology Co., Ltd., 4F, No. 2, Fuxing 4th Rd., Qianzhen District, Kaohsiung 80661, Taiwan

**Keywords:** narrow-band imaging, hyperspectral imaging, decorrelated color space, peak signal to noise ration, structural similarity index metric, entropy, YOLO

## Abstract

**Simple Summary:**

Esophageal carcinoma (EC) is a major cause of cancer deaths since it is first undetectable in its early stages. Narrow-band imaging (NBI) detects EC more accurately, sensitively, and specifically than white light imaging (WLI), according to many studies. This work uses a color space connected to décor to change WLIs into NBIs, improving early EC identification. The YOLOv5 algorithm was utilized to train WLI and NBI images separately, demonstrating the versatility of sophisticated object identification approaches in medical image analysis. Based on the confusion matrix and the trained model’s precision, recall, specificity, accuracy, and F1-score, the model’s performance was assessed. The model was trained to reliably identify dysplasia, squamous cell carcinoma (SCC), and polyps, demonstrating a detailed and focused examination of EC pathology for a better understanding. Dysplasia cancer, a pre-cancerous stage that may increase five-year survival, was detected with higher recall and accuracy by the NBI model. Although the NBI and WLI models recognized the polyp identically, the SCC category lowered its accuracy and recall rate. The NBI model had an accuracy of 0.60, 0.81, and 0.66 in dysplasia, SCC, and polyp categories, and recall rates of 0.40, 0.73, and 0.76. In dysplasia, SCC, and polyp categories, the WLI model was 0.56, 0.99, and 0.65 accurate. Additionally, it had recall rates of 0.39, 0.86, and 0.78 in the same categories. The NBI model performs poorly due to a small collection of training pictures. Increasing the dataset can improve performance.

**Abstract:**

Esophageal carcinoma (EC) is a prominent contributor to cancer-related mortality since it lacks discernible features in its first phases. Multiple studies have shown that narrow-band imaging (NBI) has superior accuracy, sensitivity, and specificity in detecting EC compared to white light imaging (WLI). Thus, this study innovatively employs a color space linked to décor to transform WLIs into NBIs, offering a novel approach to enhance the detection capabilities of EC in its early stages. In this study a total of 3415 WLI along with the corresponding 3415 simulated NBI images were used for analysis combined with the YOLOv5 algorithm to train the WLI images and the NBI images individually showcasing the adaptability of advanced object detection techniques in the context of medical image analysis. The evaluation of the model’s performance was based on the produced confusion matrix and five key metrics: precision, recall, specificity, accuracy, and F1-score of the trained model. The model underwent training to accurately identify three specific manifestations of EC, namely dysplasia, squamous cell carcinoma (SCC), and polyps demonstrates a nuanced and targeted analysis, addressing diverse aspects of EC pathology for a more comprehensive understanding. The NBI model effectively enhanced both its recall and accuracy rates in detecting dysplasia cancer, a pre-cancerous stage that might improve the overall five-year survival rate. Conversely, the SCC category decreased its accuracy and recall rate, although the NBI and WLI models performed similarly in recognizing the polyp. The NBI model demonstrated an accuracy of 0.60, 0.81, and 0.66 in the dysplasia, SCC, and polyp categories, respectively. Additionally, it attained a recall rate of 0.40, 0.73, and 0.76 in the same categories. The WLI model demonstrated an accuracy of 0.56, 0.99, and 0.65 in the dysplasia, SCC, and polyp categories, respectively. Additionally, it obtained a recall rate of 0.39, 0.86, and 0.78 in the same categories, respectively. The limited number of training photos is the reason for the suboptimal performance of the NBI model which can be improved by increasing the dataset.

## 1. Introduction

Esophageal cancer (EC) ranks sixth in terms of mortality and eighth in terms of incidence among all types of cancer [1,2,3]. Diagnosing EC in its early phases has been challenging due to the absence of distinct traits [4]. Consequently, the overall survival rate of EC has been below 20% [5,6,7]. EC ranks as the second worst kind of cancer in many nations [8]. Most patients are not eligible for curative surgical resection due to the presence of advanced illness at the time of diagnosis [9]. Early identification of EC is crucial for effective therapy and may greatly improve the five-year survival rate [10].

Several computer-aided diagnostic (CAD) systems have been developed for the detection of EC. Tsai et al. used band-selective hyperspectral imaging (HSI) to identify EC by using both the WLI and NBI images [10]. Fang et al. used image semantic segmentation, using U-net and Res-net, to accurately predict and assign labels to EC using both narrow-band images (NBI) and white-light images (WLI) [11]. Guo et al. used a SegNet architecture CAD model to perform real-time automated diagnosis of precancerous lesions using 6473 NBI images [12]. In addition to CAD, biosensors serve as an economical means for detecting cancer at an early stage [13,14,15]. Nevertheless, the capacity of these nano-materials to adapt to the environment continues to be a difficult task [16,17].

Multiple studies have shown that NBI pictures have superior performance in identifying early cancers compared to WLI images [18,19,20,21]. This is a method of imaging that uses light of certain wavelengths to enhance the endoscope and increase the visual characteristics of the mucosa [22]. The wavelengths often used are 540 nm for the green band and 415 nm for the blue band [23]. The reason for this is that the blue wavelength (415 nm) is the wavelength at which the surface mucosa absorbs most light, whereas the green wavelength (540 nm) is the wavelength at which the submucosa layer absorbs most light [24,25,26].

Many studies have previously used NBI to detect and classify EC. In a similar study by Su et al., PET/CT scan was inferior to narrow-band imaging endoscopy in detecting second primary esophageal cancer in head and neck cancer patients [27]. Another study by Yoshida et al. proved that the NBI system improved the accuracy of magnifying endoscopy for the assessment of esophageal lesion [28]. While NBI systems provide several benefits, they also come with certain drawbacks. Nevertheless, the absence of uniformity in the many NBI systems and models included in the study has rendered the assessment of the results [29] unfeasible. The role of NBI in routine clinical practice during colonoscopy is not yet well defined [30]. Effective use of NBI requires training and expertise [31]. Despite these drawbacks, NBI remains valuable, especially when combined with other modalities, aiding in lesion detection and characterization. Conventional endoscopes are the only ones that provide traditional NBI, whereas video capsule endoscopes (VCE) lack NBI capabilities due to their fixed size. The existing literature lacks a comprehensive exploration of transforming WLIs into NBIs using a color space linked to décor for the enhanced early detection of EC. Prior studies have primarily focused on the efficacy of NBI compared to WLI in detecting EC, without delving into innovative image transformation techniques. Our study bridges this gap by introducing a novel approach that leverages decor-related color spaces to transform WLIs into NBIs.

Thus, this study employed an NBI conversion technology that relies on decorrelated color space to convert WLI images into NBI images. YOLOv5 was utilized to detect and classify EC into three categories: dysplasia, SCC, and polyp. The performance of the developed model was evaluated based on five indicators: precision, recall, specificity, accuracy, and F1-score. This method significantly contributes to the existing works by offering a nuanced and targeted analysis, addressing diverse aspects of EC pathology and improving the overall understanding of its early stages.

## 2. Materials and Methods

### 2.1. Dataset

Acquiring the required information for identifying and classifying the esophagus may sometimes be a difficult task [32]. The current study used a collection of 3415 white light imaging (WLI) photographs acquired using the conventional endoscope (CV-290, Olympus Corporation, Tokyo, Japan) for analysis. The dataset including the Olympus photos was acquired at the Chung-Ho Memorial Hospital at Kaohsiung Medical University. As part of the picture preparation, each image was standardized to a fixed size of 640 × 640 pixels. This was carried out to prevent issues like limited computer capacity and to maintain a similar format across all images. LabelImg software was used, generating an XML file, which was then transformed into a text file [33]. Then, this file underwent conversion into a txt format and was then used as input for training the YOLOv5 model. It should be noted that the picture file with annotations was transformed into an NBI format before the training process. The dataset was partitioned into training, testing, and validation sets at a ratio of 70:30. The PyTorch deep learning framework was constructed on the Windows 11 operating system. The program, scripted in Python, was executed in Jupyter Notebook, utilizing the Python 3.9.18.

### 2.2. Narrow Band Imaging

The use of NBI is crucial in detecting EC at an early stage because of its superior performance compared to WLI in using CAD machine-learning approaches [33,34,35,36]. Therefore, in the current investigation, a color space that simulated the NBI picture was chosen to contain axes that were not connected. This is because it is an effective tool for altering color pictures. The implementation of the method suggested by Reinhard et al. was practically used in this context, successfully fulfilling its intended goal [37]. By including reliable input photos, the straightforward task of applying the mean and standard deviation (SD) uniformly throughout a dataset is a reasonably simple procedure that, when executed accurately, yields flawless output images. Therefore, we calculated these parameters: mean and SD by comparing the original picture with the image that would serve as the target. It is crucial to give careful consideration to the fact that we separated computations to determine the average and standard deviation of each axis in the *l* color space. The *l* axis represents an achromatic channel, while the *α* and *β* channels are chromatic yellow–blue, and red–green opponent channels. Firstly, we provide a method that, if successful, will allow the transformation of *RGB* data into *l*, the color space based on perception that was created by Ruderman et al. Since *l* is a transformation of *LMS* cone space, the first stage involves using the *LMS* transform to turn the picture into *LMS* space via a two-part process. This is essential since “*l*” is a transformation of the *LMS* cone space. The first step required is the conversion of the *RGB* tristimulus values to the XYZ tristimulus values. By obtaining a vector, we may then conduct column multiplication, yielding the desired *RGB* to XYZ conversion. This is achieved by using the conventional matrix given by the International Telecommunications Union (ITU). The image can be transformed such that it is situated in *LMS* space by making use of the usual conversion matrix, as indicated in Equation (1).
(1)$$\left(\begin{array}{c} L\\ M\\ S \end{array}\right)=\left(\left.\begin{array}{ccc} 0.3811 & 0.5783 & 0.0402\\ 0.1967 & 0.7244 & 0.0782\\ 0.0241 & 0.1288 & 0.8444 \end{array}\right).\left(\begin{array}{c} R\\ G\\ B \end{array}\right)\right.$$

By applying the logarithmic transformation described in Equation (2), the data may effectively reduce the high degree of skewness present in the existing color space.
(2)L=log⁡LM=log⁡MS=log⁡S

To start, subtract the average value from each of the data points. Subsequently, as seen in Equation (3), modifications are made to the data points that constitute the synthetic picture by using variables that are defined by the standard deviations of each individual data point. Specifically, the aim is to transfer some characteristics of the data point distribution in *lαβ* space from one image to another in a more formal manner. For this specific need, the average and variability across all of these three axes are enough. Therefore, *lαβ* metrics are calculated based on the original and target photos.
(3)$$l^{\prime }=\frac{\sigma_{t}^{l}}{\sigma_{s}^{l}}l^{*}\;\alpha^{\prime }=\frac{\sigma_{t}^{\alpha }}{\sigma_{s}^{\alpha }}\alpha^{*}\;\beta^{\prime }=\frac{\sigma_{t}^{\beta }}{\sigma_{s}^{\beta }}\beta^{*}$$

By assuming the intention to simulate the visual characteristics of one picture onto another, it is feasible to choose a source image and a target image that are not compatible. The ultimate product’s quality will depend on the extent to which the photographs possess comparable compositional aspects. If the synthetic image has a substantial proportion of grass compared to the photograph, which has a larger proportion of sky, it is logical to conclude that the process of altering the statistics will not be successful. Please refer to Appendix A for the 20 randomly selected images of WLI in VCE, and Appendix A for the 20 randomly selected images of NBI in VCE.

### 2.3. Results of the NBI Conversion

The SSIM, entropy, and PSNR comparison was conducted by comparing 20 randomly selected WLI images with their equivalent simulated NBI images [38,39]. These three characteristics provide insight into the quality of the simulated pictures. The average structural similarity index (SSIM) of Olympus photos was 98.45%. For a detailed comparison of SSIM values for each of the 20 randomly selected images in Olympus and VCE, please refer to Appendix A. Out of the 20 photographs picked at random, only four had structural similarity index (SSIM) values below 97%, while 11 images had an SSIM over 99%. These four photos were distorted due to blurring, light reflection, or flaring, resulting in a decrease in SSIM (structural similarity index measure). Improved NBI picture reproduction is achieved by using filtered datasets that have clearer WLIs. The average disparity in entropy between WLI and simulated NBI in the Olympus endoscope was 3.4552%. Please refer to Appendix A for the entropy comparison findings of each of the 20 randomly selected pictures in Olympus and VCE. The results indicate that the entropy patterns of Olympus endoscopy’s WLI pictures and NBI images are comparable. A single picture caused an increase in the NBI entropy because of an excessive amount of reflection in the WLI image. The Olympus endoscope exhibits an average peak signal-to-noise ratio (PSNR) value of 28.06 dB. For a detailed comparison of PSNR values for each of the 20 randomly selected pictures in Olympus and VCE, please refer to Appendix A. Therefore, the suggested approach demonstrated superior performance in the three selected parameters: SSIM, entropy, and PSNR. This guarantees that the NBI simulation will be precise, and independent of any picture faults. Please refer to Appendix A for the 12 randomly selected WLI photographs, simulated NBI images, and a comparable original NBI in Olympus endoscope; Olympus WLI pictures: (b) Simulated narrow-band imaging (NBI) picture and (c) a comparable NBI image captured by Olympus.

### 2.4. YOLOv5

The selection of YOLOv5 for this investigation was based on previous research that demonstrated its superior detection speed compared to other models like RetinaNet or SSD, resulting in enhanced real-time performance [40]. YOLO is specifically engineered for the purpose of detecting objects in real time [41]. Its efficient single-pass structure compared to the multiple passes of other models such as R-CNN (convolutional neural network) [42], fast R-CNN [43], and faster R-CNN [44] enables rapid picture processing. However, real-time analysis is essential in medical imaging for activities such as identifying problems during procedures or monitoring patient status. YOLO’s quick inference time makes it well suited for applications that need prompt decision making. YOLO accomplishes this by partitioning the input image into a grid and making simultaneous predictions of bounding boxes and class probabilities for each grid cell, enhancing its ability to recognize and categorize many items of interest in a single medical image streamlining the whole process, making it particularly helpful in medical imaging [45]. YOLO is renowned for its efficacy in terms of model compactness and computational demands which is of utmost importance in medical imaging applications, particularly in situations when resources are few or when there is a need to handle huge datasets rapidly. The YOLO framework benefits from a thriving open-source community, which leads to ongoing enhancements, updates, and the accessibility of pre-trained models. YOLOv5 [46] consists of three primary components: the backbone, neck, and head terminals. The backbone of Yolov5 mainly comprises models such as focus, CONV-BN-Leaky ReLU (CBL), cross-stage partial (CSP), and spatial pyramid pooling (SPP) models. For a detailed illustration of the Yolov5 architecture, please refer to Appendix A [47]. Focus can consolidate diverse detailed pictures, partition the input image, reduce the required CUDA memory and layers, enhance both the speed of forward and backward propagation, and generate image features. In this process, the input picture, with a fixed resolution of 640 × 640 pixels, is divided into four images of 320 × 320 pixels each using focus. These images are then processed by CON-CAT, along with slices and layers of convolution kernels, to generate a 320 × 320 pixel image. This process is done to accelerate the training process. The Spatial Pyramid Pooling (SPP) layer is designed to address the issue of input size restrictions while preserving the integrity of the picture. Neck refers to a series of layers that gather visual information and generate feature pyramid (FPN) and path aggregation (PAN) networks [48]. The composition includes the CBL, Upsample, CSP2_X, and other models. YOLOv5 incorporates the CSP1_X structure from the YOLOv4 version CSPDarknet-53 to reduce the size of the model and extract more complete picture characteristics. Additionally, it introduces the CSP2_X structure. The GIoU Loss is the loss function used for bounding boxes in the Head.

The loss function in YOLOv5 utilizes three types of losses: classification losses, confidence losses, and bounding box regression losses. Please refer to Appendix A for the convergence of loss functions for the training set of WLI images, as well as precision, recall, and average precision. Additionally, Appendix A provides information on the loss functions and convergence of precision, recall, and mean precision for the NBI image training set [49,50]. The loss function quantifies the disparity between the actual and anticipated values of the model. The loss function of the YOLOv5 model is expressed as follows:(4)$$L_{GIOU}=\sum_{i=0}^{S^{2}}\sum_{j=0}^{B}I_{ij}^{obj}\left[1-IOU+\frac{A^{c}-U}{A^{c}}\right]$$
where *B* is the number of bounding boxes in each grid and *S*^2^ is the total number of grids. When an object is present in the bounding box, the value of $I_{ij}^{obj}$ is equal to 1, otherwise it is equal to 0.
(5)−λnoobj∑i=0S2∑j=0BIijnoobj⌢jCilogCij+1−Cijlog1−Cij
where ⌢jCi is the actual confidence of the bounding box of *j* in the grid of *i*, $C_{i}^{j}$ is the predicted confidence of the bounding box of *j* in the grid of *i*, and $\lambda_{noobj}$ is the confidence weight when objects are missing from the bounding box.
(6)$$L_{class}=-\sum_{i=0}^{S^{2}}I_{ij}^{noobj}$$
(7)∑c∈classes⌢jPiclogPijc+1−⌢jPiclog1−Pijc
where ⌢jPic is the predicted likelihood that the detected object will fall under the category, and $P_{i}^{j}\left(c\right)$ is the likelihood that it actually does.

## 3. Results

The system was programmed to identify three distinct cancer conditions: dysplasia, SCC, and esophageal polyp. In the context of dysplasia, the precancerous stage is characterized by the development of aberrant cellular features, but these cells are not yet capable of spreading to other areas [51]. SCC is an often lethal disease that commonly presents as increasing difficulty in swallowing for elderly individuals [52]. Esophageal polyps are a rare kind of benign esophageal cancer composed of adipose tissue, veins, and fibrous tissue [53]. Figure 1 displays the forecast outcomes of the EC in the WLI, whereas Figure 2 exhibits the forecast outcomes of the EC in the NBI picture. The research uses label A to indicate dysplasia, label B to represent SCC, and label C to represent polyp.

Table 1 presents a comparison of the predictive accuracy of the WLI in relation to NBI pictures. For the precision–confidence curve of the WLI image dataset, refer to Appendix A. For the precision–confidence curve of the NBI image dataset, refer to Appendix A. The performance results were evaluated by analyzing many criteria, such as precision, recall, specificity, accuracy, F1-score, and mAP50. The use of NBI pictures has resulted in a noticeable improvement in accuracy and recall. Specifically, the precision has grown from 0.56 to 0.60, and the recall has increased from 0.39 to 0.40 when comparing NBI photos to WLI images. Nevertheless, there is a notable decrease in both accuracy and recall when identifying SCC. Specifically, the precision drops from 0.99 to 0.81 and the recall drops from 0.86 to 0.73 when transitioning from WLI photos to NBI images. When it comes to detecting polyps, both WLI and NBI showed comparable performance. The accuracy for WLI and NBI was 0.65, while the recall was 0.78. In comparison, the precision for WLI was 0.66 and the recall was 0.76.

The dysplasia category in the NBI model has a much greater enhancement in comparison to the WLI model. Dysplasia is a condition that precedes cancer, and promptly detecting dysplasia may significantly enhance the overall five-year survival probability of EC. This discovery demonstrated the model’s exceptional level of comprehensiveness in acquiring lesion features. Due to the little quantity of photos needed for training the WLI and NBI models, it is evident that their accuracy rates are almost comparable. Nevertheless, the findings of this work demonstrate that decorrelated axes-based conversion algorithms, including the capability to transform WLI pictures into NBI images, have promise for classifying and recognizing EC. This deduction may be made based on the fact that these algorithms have the capability to transform WLI photographs into NBI images. This study has significantly improved both the ability to correctly identify cases of dysplasia and the accuracy in classifying them. This research has improved the specificity of the polyp category.

## 4. Discussion

The study’s findings suggest that the NBI model can greatly enhance the accuracy of identifying dysplasia and polyps, the two categories of EC. In contrast, the NBI model diminishes the accuracy of the SCC category. This is due to the limited number of photos available in the datasets used for this investigation. Nevertheless, the accuracy can be substantially enhanced by augmenting the dataset with a greater number of SCC cases. The upcoming stage of this research project will focus on implementing the decorrelated axis conversion technology, which enables the conversion of WLI images to NBI images. These converted images will subsequently be employed in a video capsule endoscopy (VCE). Due to spatial constraints in VCE, it is not feasible to incorporate an additional NBI filter. This would enhance the device’s capacity to identify and classify various forms of cancer. Simply applying the identical procedure suffices to convert WLI photos to NBI images in VCE. Another advantage of employing this approach is its ability to decrease the cost of the NBI acquisition system. Therefore, this technique can be utilized on the previously obtained WLI images to diagnose different types of cancer. This approach enables the automatic and real-time diagnosis of many types of cancer. Currently, this approach is not employed in any clinical diagnosis. However, if further research is conducted on converting WLI images to NBI for esophageal cancer, employing this method in conjunction with the detection and diagnosis methods of this study will be beneficial for early EC diagnosis and to enhance predictive accuracy. In this study, the focus was on specific aspects within the realm of esophageal conditions, highlighting certain dysplastic states while acknowledging there are broader conditions such as Barrett’s esophagus that are pertinent to this field, which is the future scope of this study. The combination of high-grade and low-grade dysplasia could be a point of contention. However, this decision to amalgamate these dysplastic states was to streamline the discussion and emphasize the broader trends in dysplastic changes observed in various conditions. However, in the future, separately delving into the nuances and differences between high-grade and low-grade dysplasia is necessary. In order to address possible sources of bias or confounding variables in future research, it is necessary to use a multi-center strategy that includes a wide range of demographic populations and medical institutions. To maximize the generalizability of the findings, it is important to have a well-proportioned representation of age, gender, and geographical areas in the dataset. Furthermore, the utilization of several object identification algorithms in conjunction with YOLO architectures and the subsequent comparison of their performances might yield a more thorough comprehension of the model’s capabilities and limitations. Standardizing imaging methods and implementing quality assurance measures across several sites helps minimize variability in picture features. In light of the current study’s focus on YOLO for medical applications, the future research endeavors will extend to exploring the efficiency of alternative object detection algorithms, including R-CNN, fast R-CNN, and faster R-CNN. Recognizing the significance of benchmarking and comparing various models, the aim would be to conduct a comprehensive performance analysis to assess their suitability for specific tasks within the realm of medical imaging. This exploration will provide valuable insights into the strengths and limitations of different architectures, contributing to the ongoing refinement of methodologies. Cooperative endeavors and the exchange of datasets across researchers can help to address biases and improve the reliability of medical image analysis algorithms.

## 5. Conclusions

In this research, a decorrelated color space was employed to convert WLI photos into NBI images, and a YOLOv5 model was used to detect and classify EC. The model underwent training to accurately identify three specific types of EC: dysplasia, SCC, and polyps. The NBI model effectively enhanced both its recall and precision rates in detecting dysplasia cancer, a precancerous stage that has the potential to enhance the overall five-year survival rate. Conversely, the SCC category had a decrease in both its precision and recall rate. This is due to the limited quantity of photos utilized for training the model. Enhancing the NBI model’s performance can be achieved by augmenting the training dataset with a larger number of photos. In subsequent instances, an identical approach can be employed to transform the WLI images derived from the VCE into NBI images. Subsequently, these photos can be merged with a suitable artificial intelligence model to identify and categorize various forms of cancer.

## Figures and Tables

**Figure 1 cancers-16-00572-f001:**
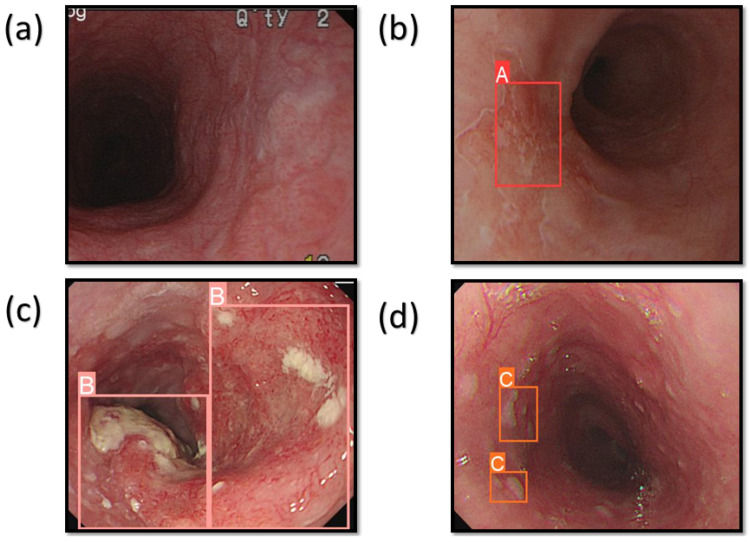
The results of the prediction of EC on the WLI images. (**a**) When the condition of the EC is normal; (**b**) represents the dysplasia with the red bounding box surrounding the lesion area with the text A; (**c**) shows the area with SCC and the pink bounding box surrounding the lesion area with the text B; (**d**) shows the area of the polyp with the orange bounding box surrounding the lesion area with the text C.

**Figure 2 cancers-16-00572-f002:**
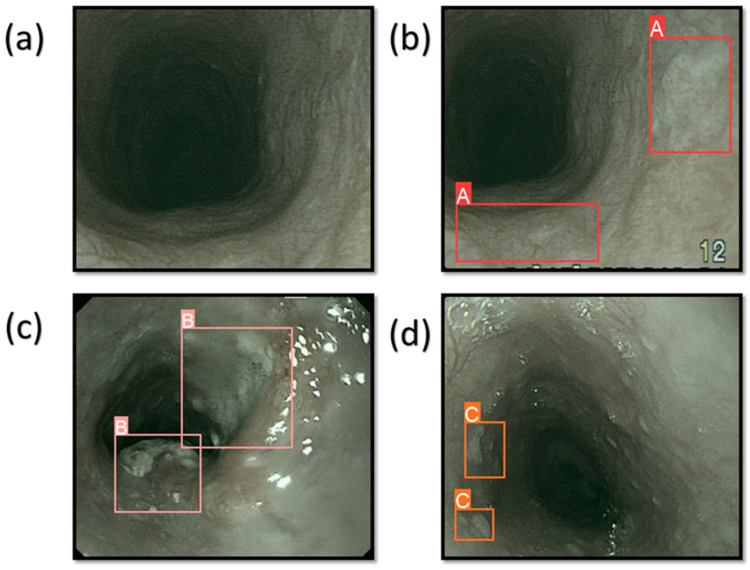
The results of the prediction of EC on the NBI images. (**a**) When the condition of the EC is normal; (**b**) represents the dysplasia with the red bounding box surrounding the lesion area with the text A; (**c**) shows the area with SCC and the pink bounding box surrounding the lesion area with the text B; (**d**) shows the area of the polyp with the orange bounding box surrounding the lesion area with the text C.

**Table 1 cancers-16-00572-t001:** The performance results of YOLOv5 training.

WLI	Precision	Recall	F1-score	mAP50
Dysplasia	0.56	0.39	0.46	0.41
SCC	0.99	0.86	0.92	0.933
Polyp	0.65	0.78	0.71	0.79
**NBI**	**Precision**	**Recall**	**F1-score**	**mAP50**
Dysplasia	0.60	0.40	0.47	0.42
SCC	0.81	0.73	0.77	0.82
Polyp	0.66	0.76	0.71	0.78

## Data Availability

The data presented in this study are available in this article upon request to the corresponding author.

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
