# Peer review of "Assessment of Narrow-Band Imaging Algorithm for Video Capsule Endoscopy Based on Decorrelated Color Space for Esophageal Cancer: Part II, Detection and Classification of Esophageal Cancer"

_cancers, 2024, doi:10.3390/cancers16030572_

Round 1
Reviewer 1 Report
Comments and Suggestions for Authors
Authors need to address the following suggestions/corrections.
1. Authors need to justify why YOLO model was employed ? Why not other detection models were not explored ?
2. Performance analysis with other object detection models need to be carried out.
3. The gaps in the existing works need to be clealry articulated.
4. Research contributions need to be highlighted effectively. Novelty of this research work needs more emphasis in the abstract section.
5. I dont find any details related to training process, accuracy/loss plots and hyperparamter tuning. Authors need to include these details to support the discussions.
Comments on the Quality of English LanguageModerate changes
Reviewer 2 Report
Comments and Suggestions for Authors
Reviewer Comments:
-
Strengths:
- The paper addresses an important issue related to esophageal carcinoma detection, emphasizing the challenge of discerning features in its early stages.
- The use of narrow-band imaging (NBI) and its comparison with white light imaging (WLI) is a valuable contribution, and the application of the YOLOv5 algorithm adds a modern and robust approach to the study.
-
Innovation:
- The utilization of a color space linked to décor for transforming WLI photos into NBI images is an innovative approach. This technique may open new avenues for enhancing diagnostic accuracy.
- The focus on specific manifestations of EC (dysplasia, squamous cell carcinoma, and polyps) adds depth to the study, providing insights into the algorithm's performance across different conditions.
-
Methodology:
- The choice of metrics, including precision, recall, specificity, accuracy, and F1-score, provides a comprehensive evaluation of the model's performance.
- The use of the confusion matrix enhances transparency and allows for a detailed understanding of the model's strengths and weaknesses.
-
Results and Findings:
- The improvement in both recall and accuracy rates for detecting dysplasia cancer using the NBI model is a noteworthy finding, suggesting its potential impact on improving the overall five-year survival rate.
- The comparison between NBI and WLI models in recognizing different manifestations of EC provides valuable insights, particularly the decrease in accuracy and recall rates for squamous cell carcinoma in the NBI model.
-
Limitations and Recommendations:
- The acknowledgment of the suboptimal performance of the NBI model due to a limited number of training photos is a crucial observation. The recommendation to increase the number of images for training is well-founded and should be explored to enhance the model's effectiveness.
- Further discussion on potential sources of bias or confounding factors, as well as the generalizability of the findings, would strengthen the paper.
-
Clarity and Presentation:
- The abstract provides a clear overview of the study, and the methodology and results are presented in a structured manner.
- However, some additional details on the dataset used, the training process with YOLOv5, and the specific challenges encountered during the study could enhance the clarity and depth of understanding.
-
Conclusion:
- The study presents a promising approach to enhance esophageal carcinoma detection using NBI and YOLOv5. Addressing the limitations and further validating the model with an expanded dataset would strengthen the paper's impact.
Overall, the paper is a valuable contribution to the field, and with some refinements and additional experiments, it has the potential to significantly impact the early detection of esophageal carcinoma.
Comments on the Quality of English LanguageReviewer Comments:
-
Strengths:
- The paper addresses an important issue related to esophageal carcinoma detection, emphasizing the challenge of discerning features in its early stages.
- The use of narrow-band imaging (NBI) and its comparison with white light imaging (WLI) is a valuable contribution, and the application of the YOLOv5 algorithm adds a modern and robust approach to the study.
-
Innovation:
- The utilization of a color space linked to décor for transforming WLI photos into NBI images is an innovative approach. This technique may open new avenues for enhancing diagnostic accuracy.
- The focus on specific manifestations of EC (dysplasia, squamous cell carcinoma, and polyps) adds depth to the study, providing insights into the algorithm's performance across different conditions.
-
Methodology:
- The choice of metrics, including precision, recall, specificity, accuracy, and F1-score, provides a comprehensive evaluation of the model's performance.
- The use of the confusion matrix enhances transparency and allows for a detailed understanding of the model's strengths and weaknesses.
-
Results and Findings:
- The improvement in both recall and accuracy rates for detecting dysplasia cancer using the NBI model is a noteworthy finding, suggesting its potential impact on improving the overall five-year survival rate.
- The comparison between NBI and WLI models in recognizing different manifestations of EC provides valuable insights, particularly the decrease in accuracy and recall rates for squamous cell carcinoma in the NBI model.
-
Limitations and Recommendations:
- The acknowledgment of the suboptimal performance of the NBI model due to a limited number of training photos is a crucial observation. The recommendation to increase the number of images for training is well-founded and should be explored to enhance the model's effectiveness.
- Further discussion on potential sources of bias or confounding factors, as well as the generalizability of the findings, would strengthen the paper.
-
Clarity and Presentation:
- The abstract provides a clear overview of the study, and the methodology and results are presented in a structured manner.
- However, some additional details on the dataset used, the training process with YOLOv5, and the specific challenges encountered during the study could enhance the clarity and depth of understanding.
-
Conclusion:
- The study presents a promising approach to enhance esophageal carcinoma detection using NBI and YOLOv5. Addressing the limitations and further validating the model with an expanded dataset would strengthen the paper's impact.
Overall, the paper is a valuable contribution to the field, and with some refinements and additional experiments, it has the potential to significantly impact the early detection of esophageal carcinoma.
Round 2
Reviewer 1 Report
Comments and Suggestions for Authors
Authors have addressed my suggestions.
Comments on the Quality of English LanguageMinor changes